# Bovine Enteroids as an In Vitro Model for Infection with Bovine Coronavirus

**DOI:** 10.3390/v15030635

**Published:** 2023-02-27

**Authors:** Ruchika Shakya, Alejandro Jiménez-Meléndez, Lucy J. Robertson, Mette Myrmel

**Affiliations:** Department of Paraclinical Sciences, Faculty of Veterinary Medicine, Norwegian University of Life Sciences (NMBU), 1430 Ås, Norway

**Keywords:** bovine coronavirus (BCoV), bovine, enteroid, in vitro model, organoid

## Abstract

Bovine coronavirus (BCoV) is one of the major viral pathogens of cattle, responsible for economic losses and causing a substantial impact on animal welfare. Several in vitro 2D models have been used to investigate BCoV infection and its pathogenesis. However, 3D enteroids are likely to be a better model with which to investigate host–pathogen interactions. This study established bovine enteroids as an in vitro replication system for BCoV, and we compared the expression of selected genes during the BCoV infection of the enteroids with the expression previously described in HCT-8 cells. The enteroids were successfully established from bovine ileum and permissive to BCoV, as shown by a seven-fold increase in viral RNA after 72 h. Immunostaining of differentiation markers showed a mixed population of differentiated cells. Gene expression ratios at 72 h showed that pro-inflammatory responses such as IL-8 and IL-1A remained unchanged in response to BCoV infection. Expression of other immune genes, including CXCL-3, MMP13, and TNF-α, was significantly downregulated. This study shows that the bovine enteroids had a differentiated cell population and were permissive to BCoV. Further studies are necessary for a comparative analysis to determine whether enteroids are suitable in vitro models to study host responses during BCoV infection.

## 1. Introduction

Bovine coronavirus (BCoV) is one of the major respiratory and enteric viral pathogens of cattle, causing a substantial impact on animal welfare in the beef and dairy industries [1,2,3]. BCoVs, belonging to the family *Coronaviridae*, genus *Betacoronavirus*, are enveloped, positive-sense, single-stranded RNA (+ssRNA) viruses with a genome of 31 Kb and a diameter of 120–160 nm [4]. The virus is globally distributed [5] and endemic in Scandinavian countries, including Norway. BCoV is transmitted by inhalation of aerosols from nasal discharge, thereby reaching the respiratory epithelium, or reaching the intestine via the faeco-oral route [6]. BCoV is the cause of three distinct clinical syndromes in cattle—calf diarrhoea (CD), winter dysentery (WD) with haemorrhagic diarrhoea in adults, and respiratory infections in cattle of various ages—and is an important contributor to bovine respiratory disease complex [5,7]. The virus is thus able to replicate both in the gastrointestinal and upper respiratory epithelium [8].

Two-dimensional (2D) cell-culture systems have been used to culture BCoV and to study host–pathogen interactions [9] and pathogen–pathogen interactions [10]. The cell lines used to date include human rectal tumour-18 (HRT-18G), human colon adenocarcinoma (HCT-8), African green monkey kidney, Madin Darby bovine kidney, Madin Darby canine kidney 1, bovine embryonic lung, bovine embryonic kidney, and bovine foetal spleen cell lines [11]. Although cell lines are useful experimental models, they lack cellular diversity, and functionality, and some may present genomic abnormalities [12]. On the other hand, primary bovine intestinal epithelial cell lines show a better resemblance to in vivo models. However, they are not well characterized, proliferate slowly, and can be used for only a limited number of passages [13]. Finally, more optimal in vivo animal studies are more expensive, require large numbers of animals and the necessary facilities, and their use has ethical implications.

To overcome some of the limitations associated with cell lines or animal models, organoids, multicellular three-dimensional (3D) in vitro models exhibiting similar architecture as organs [14], could be a good alternative. Organoids can be grown in an extracellular matrix (ECM) supplemented with several growth factors, allowing long-term propagation [15]. Enteroids are organoids derived from the intestine; they form an enclosed system, with the basal regions of the epithelium facing outwards and apical brush borders lying inwards [16,17,18]. Enteroids comprise a mixed population of stem cells, enterocytes, Paneth cells, goblet cells, and enteroendocrine cells, and can be used to study host interactions with enteric pathogens [19,20,21].

Recently there has been an upsurge in studies focusing on enteroids as suitable systems for culturing pathogens [22,23,24]. Human intestinal organoids have proven successful for culturing human rotavirus (HuRoV), norovirus, and enteroviruses [22,25,26], and porcine enteroids have been used to study infection caused by porcine enteric coronaviruses and the host innate response [27]. Although studies on bovine enteroids infected with enteric pathogens are limited to date, there are some studies on the characterization of cell types and culture of bovine rotavirus (BRoV) group A, Salmonella typhimurium, Toxoplasma gondii, and Mycobacterium avium subspecies paratuberculosis [17,28,29].

The aim of the present study was to establish bovine enteroids as a replication system for BCoV, to characterize the enteroid cell population, and to compare the expression of selected immune genes during BCoV infection with that previously described in BCoV-infected HCT-8 cells [30]. The gene expression of intestinal stem cells (LGR5+) and differentiation markers (ChrA and Muc2) were analysed using qPCR followed by immunostaining to look for proteins (ChrA, Muc2, sucrase isomaltase, and lysozyme). A panel of three commonly used housekeeping genes (18s rRNA, GAPDH, and ACTB) were tested [31,32,33], to normalize the gene expression.

## 2. Materials and Methods

### 2.1. Isolation of Intestinal Crypts

Tissues used were obtained from post-mortem of a healthy male British Holstein–Friesian (*Bos taurus*) calf less than 1 month old. The isolation of intestinal crypts was carried out at Moredun Research Institute (MRI), Scotland, UK as described in [18]. For all the animal studies carried out at MRI, regulatory licenses had been approved by the University of Edinburgh’s Ethics Committee (E50/15).

Briefly, a 10 cm portion of the ileum was collected into ice-cold phosphate-buffered saline (PBS; Gibco, Thermo Fisher Scientific, Scotland, UK) containing 25 μg/mL gentamicin (gen) and 100 U/mL penicillin/streptomycin (pen/str) (Life Technologies, Paisley, UK). The ileum was cut open longitudinally and a glass slide was used to gently remove the mucus layer. The mucosal layer consisting of the intestinal stem cells were then scraped off and suspended in 15 mL of Hank’s Balanced Salt Solution (HBSS; Gibco, Thermo Fisher Scientific) with 25 μg/mL gen and 100 U/mL pen/str. The sample was washed several times with HBSS and digested at 37 °C for 40 min in 25 mL of Dulbecco’s Modified Eagle’s Medium (DMEM; Gibco, Thermo Fisher Scientific) containing 1.0% foetal calf serum (Thermo Fisher Scientific, Grand Island, NY, USA), 25 μg/mL gen, 100 U/mL pen/str, 75 U/mL collagenase type1-A (Sigma-Aldrich, St. Louis, MO, USA) and 20 μg/mL dispase I (Roche, Mannheim, Germany). The number and integrity (finger-like morphology) of isolated crypts were visually assessed by light microscopy at 4× magnification. The crypts were further washed to remove traces of digestion medium and centrifuged at 400× *g* for 2 min, before finally being re-suspended in 100 μL advanced DMEM/F12 medium (Adv F12; Thermo Fisher Scientific) with 1× B27 supplement minus vitamin A (Thermo Fisher Scientific), 25 μg/mL gen, and 100 U/mL pen/str. After the final wash, the sample consisted of 1000–2000 crypts.

### 2.2. Enteroid Culture and Differentiation

The enteroids were cultured as previously described [18] and, for long-term storage, enteroids were cryopreserved in Cryostor CS10 medium (STEMCELL Technologies, Cambridge, UK) in liquid nitrogen. Cryopreserved enteroids were resuscitated and passaged at least thrice (P3) before any experimental use in our laboratory.

The crypts (roughly 1000 crypts in 100 μL Adv F12) were added to 150 μL of Corning Growth Factor Reduced Matrigel Matrix (AH Diagnostics, Oslo, Norway). Then, 50 μL of the mix (~200 crypts) was added to each of five wells in a pre-warmed 24-well plate (Nunc, Thermo Fisher Scientific). The Matrigel polymerized for 10 min in a humidified incubator with 5% CO_2_ at 37 °C before 500 μL of growth medium was added to each well. The growth medium consisted of IntestiCult Organoid Growth Medium (Mouse), with Rho-associated kinase inhibitor (Y-27632; Cayman chemicals, Ann Arbor, MI, USA), p38 mitogen-activated protein kinase inhibitor (SB202190; Enzo Life Science, England, UK), and TGFβ inhibitor (LY2157299; Cayman chemicals), referred to as medium 1 hereafter (Appendix A) [34,35]. Every 2–3 days, the medium was replaced with fresh medium. The enlarged and budding crypts (henceforth referred to as enteroids) were passaged every 7–10 days by removing the medium and dissolving the Matrigel in 500 μL of cold Adv F12 with 1× B27 supplement minus vitamin A, 25 μg/mL gen, and 100 U/mL pen/str. The enteroids were fragmented into crypts by pipetting, counted, and plated as previously described [18].

A total of four media (Appendix A) were prepared to be tested for enteroid differentiation by immunostaining, as described later. Enteroids were grown in medium 1 for 2–3 days and the medium was replaced with 300 μL of differentiation medium (1–4).

### 2.3. Inoculation of Crypts with Bovine Coronavirus

The BCoV strain (GenBank accession ID OQ507475) originated from a calf faecal sample and had previously been isolated in our lab using the human rectal tumour cell line (HRT-18G, ATCC CRL-11663) [36]. The isolate was titrated on the cells grown in a 96-well plate by ten-fold serial dilutions, using the Spearman–Karber method [37] (TCID50/mL = 3.16 × 10^6^).

Inoculation was performed using enteroids that were: A) fragmented into crypts with pipetting only, and B) further dissociated by TrypLE Express enzyme, as detailed below.

Setup A: Approximately 1000 crypts in 100 µL were inoculated with 100 µL of BCoV (6.32 × 10^3^ TCID50) per well in a 24-well plate. After 1 h incubation at 37 °C, the crypts were transferred into an Eppendorf tube and centrifuged at 300× *g* for 5 min. The pellet was washed once, and the crypts resuspended in 100 μL Adv F12. After counting, the crypts were mixed with Matrigel, and medium 1 was added after polymerization as described for the enteroid culture.

Setup B: Approximately 1000 crypts in 500 µL that had already been fragmented by pipetting were further dissociated by incubation with 200 µL of 1× TrypLE Express enzyme (Thermo Fisher Scientific, Newton Drive, Carlsbad, CA, United States) for 10 min at 37 °C, before resuspension in Adv F12 with 300 µL of 2% foetal bovine serum (FBS; Thermo Fisher Scientific) to stop the dissociation process. The pellet was washed once, and the crypts resuspended in 100 μL Adv F12. The dissociated crypts were inoculated with same amount of BCoV and grown as described for setup A.

Three individual trials with three replicates were completed for both setups.

Samples and negative controls (mock-infected with Adv F12) were harvested after 1 h and 72 h post-inoculation (hpi), to be checked for virus replication. Prior to harvesting, the crypts were inspected by light microscopy for morphological changes. For harvest, the Matrigel was solubilized in cold Adv F12 and the crypts were centrifuged at 300× *g* for 5 min, washed, centrifuged once more, and lysed with 300 μL of RLT buffer plus DTT (2M) (Qiagen, Hilden, Germany). Investigation of virus replication was carried out using immunostaining and RT-qPCR [38].

### 2.4. Immunostaining for Cell Differentiation Markers and BCoV S-Protein

Enteroids were stained for differentiation markers after 2–3 days in medium 1–4, while the infected enteroids were stained for BCoV envelope S-protein and enteroendocrine cells at 72 hpi. The enteroids were rinsed with ice-cold PBS for the removal of Matrigel followed by a fixation and permeabilization step for 20 min as described in [10]. The blocking was performed for 1 h in PBS with 20% horse serum (Thermo Fisher Scientific), and 0.1% Tween-20. The stains and antibodies (Table 1) were diluted in PBS with 2% horse serum and 0.1% Tween 20 and incubated with the enteroids for 1 h at room temperature.

A washing step with cold washing buffer and centrifugation was incorporated twice after each incubation step.

After the final centrifugation, the pellet was resuspended in Fluoroshield (Merck) and aliquots of 20 μL were mounted on glass slides with coverslips. Preparations were examined using a Leica Inverted Confocal SP8 equipped with a White Light Laser, a Leica HyD Detector, and the Leica Application Suite software (Leica microsystems GmbH, Mannheim, Germany), and images were captured at 40× and 63× magnification under oil immersion. Counting mode was used in the sequential setting, displaying the image based on the number of photons detected per pixel over a constant integration time.

### 2.5. RNA Isolation and RT-qPCR for Detection of Viral RNA

The enteroids were harvested in 300 μL of RLT buffer plus DTT (2M) Qiagen as described in Section 2.3 and mixed well by pipetting. Total RNA was isolated from the enteroids using RNeasy Kit (Qiagen), following the manufacturer’s instructions. The RNA was eluted in 50 μL of nuclease-free water and stored at −80 °C until analysis.

Information on primers, probe, and RT-qPCR conditions is provided in Appendix A. RT-qPCR for BCoV was performed based on a published protocol [38] using the RNA UltraSense™ One-Step Quantitative RT-PCR System kit (Invitrogen, Waltham, MA, USA). A total volume of 20 μL with 2 μL of template RNA was run in a Stratagene AriaMx Real-Time PCR System (Agilent Technologies, Inc., Santa Clara, CA, USA), with Agilent Aria Software v1.5. All samples were run in technical duplicates, with negative and positive controls included in each run.

### 2.6. Relative Quantification of BCoV RNA and Statistical Analysis

Relative quantification of BCoV genome copies was performed using the formula Ns1 = Ns2*(1 + E) ^(Cts2 − Cts1)^ [39] where Ns1 and Ns2 represent sample copy numbers, Cts1 and Cts2 are sample Ct values, and E the efficiency of the RT-qPCR. Based on a standard curve prepared from a 10-fold dilution series of BCoV RNA, the efficiency was determined to be 98%. To find any statistically significant difference between copy numbers at 1 and 72 hpi, a one-way nonparametric Mann–Whitney *U*-test for independent samples was used.

### 2.7. Estimation of Gene Expression Ratios

The immune genes included in the present study were selected as they had been shown to be differentially expressed with a high fold change at 72 hpi during BCoV infection of HCT-8 cells [30]. A panel of three commonly used housekeeping genes (18s rRNA, GAPDH, and ACTB) were tested [31,32,33], and the most stably expressed (similar Ct-values in mock- and BCoV-infected enteroids) were included to normalize the gene expression results. All primers are listed in Appendix A [40,41,42,43].

The RNA used for quantification of viral RNA was treated with DNAse I (Qiagen) at room temperature (20–25 °C) for 10 min including a final DNAse heat inactivation step at 95 °C for 5 min. cDNA was synthesized from 24 µL RNA using SuperScript™ IV VILO™ Master Mix (Invitrogen, Paisley, UK) in a 48 μL volume, according to the manufacturer’s instructions.

The qPCR was performed in 20 μL volumes using PowerUp™ SYBR™ Green Master Mix (Applied Biosystems, Foster City, CA, USA), 10 pmol each of forward and reverse primer and 2 μL cDNA diluted 1:2. The qPCR was carried out using a Stratagene AriaMx Real-Time PCR instrument and the cycling steps were 2 min at 95 °C followed by 40 cycles of 15 s at 95 °C, 30 s at 60 °C, and a melting curve stage. The amplicon specificity was verified by analysing the melting curve with Agilent Aria Software v1.5. All samples were run in technical triplicates. The qPCR efficiencies (E) were determined by using 4-fold serial dilutions of cDNA and were between 94% and 110%.

A replicate *C_t_*-value that differed by more than 1 from the average of the two others was considered an outlier and omitted. Gene expression ratios were calculated using the Pfaffl method [44] with normalization against the housekeeping genes.
$$Relative\;gene\;expression\;ratio=\frac{{\left(E_{target}\right)}^{∆C_{t}\;target}}{GeoMean[{\left(E_{ref}\right)}^{∆C_{t}\;ref}]\;}$$
where *E* is the efficiency of the qPCRs, ∆*C_t_* is the difference between average *C_t_*-values of the mock-infected and BCoV-infected samples, and *GeoMean* is the geometric mean [45].

To test the statistical significance of gene expression differences between mock- and BCoV-infected enteroids at 72 hpi, average *C_t_*-values from all experiments were assessed in group means using pair-wise fixed reallocation randomization test with at least 2000 randomizations performed in REST software 2009 [46]. The criteria for considering a gene differentially expressed were ratios >1 (upregulated) or <1 (down-regulated), and a *p*-value < 0.05.

## 3. Results

### 3.1. Culture, Maintenance, and Staining for Cell Markers

The enteroids were successfully cultured in medium 1, cryopreserved, and resuscitated for experimental use. The crypts formed spheroid-like enteroids after 1–2 days and started budding out, giving enlarged enteroids with a lumen by 5–10 days (Appendix A). The enteroids could be maintained for at least 35 passages, without any obvious signs of degeneration. Staining with Phalloidin showed F-actin rich brush borders on the luminal surface (Figure 1a). The enteroids stained for sucrase isomaltase and ChrA revealed the presence of enterocytes (Figure 1b) and enteroendocrine cells (Figure 1c). However, staining for Muc2 (goblet cells) and lysozyme (Paneth cells) gave only background signals (not shown).

The enteroids that were grown in medium 1 before receiving differentiation medium 2–4 (Appendix A) survived for only 3–4 days before starting to disintegrate and die (Appendix A) and were not further used.

### 3.2. BCoV Replication

Regarding the preparation of enteroids for virus inoculation, Setup A gave finger-like crypts surrounded by some cellular debris, while Setup B resulted in smaller disintegrated crypts with small openings and more cellular debris (Appendix A).

The mock- and BCoV-inoculated crypts were morphologically intact, with no cellular debris at 1 and 72 hpi. At 72 hpi, crypts had started budding out, forming lumen-like structures (spheroids), which stained positive for BCoV-S protein (inoculated crypts from setup B). Nevertheless, relatively few infected cells were identified (Figure 1d), although the samples showed a statistically significant increase in the virus gene copy number from 1 to 72 hpi (Mann–Whitney *U* = 14, *p* < 10^−5^) (Table 2).

### 3.3. Gene Expression

For the gene expression, three trials with mock-infected and two of the inoculated enteroids were included in the analysis. The first trial with inoculated samples was excluded as the cDNA concentrations were out of range and the Ct values very high (Appendix A).

Among the three housekeeping genes tested, GAPDH and ACTB were included for the normalization of the expression results, as they were more stably expressed than 18 s rRNA (Table 3).

The differentiation markers, LGR5+ (stemness) and ChrA (enteroendocrine cells), were upregulated, while the Muc2 (goblet cells) gene was downregulated in the BCoV-infected enteroids compared to the mock-infected at 72 hpi.

The expression ratios for the immune genes, IL-8 and IL-1A, were similar for mock- and BCoV-infected enteroids, while MMP13, CXCL-3, and TNF-α genes were downregulated.

## 4. Discussion

The main finding of this study is that bovine enteroids were able to support the replication of BCoV, as indicated by an approximately seven-fold increase in viral RNA after 72 h and the immunostaining of BCoV envelope S-protein. According to our knowledge, this is the first report on BCoV replication in enteroids.

The enteroids were successfully grown from intestinal tissue (ileum) from a healthy young calf less than one month old and could be propagated in medium 1 for at least 35 passages without any signs of degeneration, which is longer than reported in other studies [17,18]. Ref. [18] also isolated crypts from the ileum of young calves (<one month old), but the enteroids could be passaged in medium 1 no more than five to eight times without any sign of degeneration. This indicates that the possible number of passages of bovine enteroids could depend on the donor animal. The region of intestine used, animal age, and breed might also have an influence, as [17] isolated crypts from the jejunum of adults (20 to 30 months) of a different breed and the enteroids could be kept in medium 2 for no more than four to 12 passages before degeneration. In our hands, medium 2 did not support the growth and differentiation of the enteroids.

Staining of enteroids in medium 1 for F-actin rich brush borders on the luminal surface indicated that the epithelium was polarised, as in vivo. The stained differentiation markers revealed enterocytes and enteroendocrine cells, while staining for goblet cells and Paneth cells gave largely non-specific signals. In total, the staining results indicate that medium 1 can support differentiation to some extent. Previous studies using bovine, porcine, and murine enteroids [17,47,48] have indicated that WntCM, growth factors R-spondin (Wnt agonist), and Noggin (TGFβ superfamily inactivator) are core components for differentiation. Ref [17] demonstrated a differentiated cell population by immunostaining (enteroendocrine and goblet cells) and by proteomic analysis (enterocytes, stem cell markers, mucus components, and villus morphogenesis). Furthermore, EGF (epidermal growth factor) is also considered crucial for the growth and differentiation of the cells. Refs [16,49] also describe the need of developing in-house media to replace IntestiCult to adjust individual growth factors. Bovine supplements/growth factors are, however, not readily available and, therefore, those derived from humans/mice are mostly used. This could be a reason for the difficulty in achieving differentiated bovine enteroids [47,50], as the homology/activity of EGF varies between species [21]. In the present study, several formulations (Appendix A) for the manipulation of growth factors and differentiation of the enteroids were tested, but our results did not indicate that modifying the concentration of growth factors mentioned above was necessary for differentiation. Embedding the enteroids in Matrigel could also have contributed to differentiation, as described by [51,52]. To lower any Matrigel-driven differentiation, we used Matrigel with reduced growth factors [18].

In addition to the media 1 and 2, as described by [17,18], we tried two other formulations. Medium 3 was tested as [21] found it to work well for chicken enteroids, which they reported to be the most difficult to grow and differentiate. Ref [34] was able to differentiate human enteroids using medium 4, and to infect them with HuRoV, while [25] also followed the same recipe for human enteroids and found them permissive for human norovirus. Ref [28] also used medium 4 to differentiate bovine enteroids and infect them with BRoV. Nevertheless, of the three formulations in Appendix A, only [18] supported persistent growth and some differentiation of bovine enteroids in our study.

The enteroids inoculated with BCoV showed no apparent signs of degeneration after 72 hpi. When the enteroids were mechanically disrupted and exposed to BCoV (setup A), a small increase in viral RNA could be detected, indicating that a limited infection had been established. However, when Trypl Xpress was used for further dissociation (setup B), there was probably an increased exposure of the interior (luminal) surface to virus particles, thus resulting in a clearly recognizable infection despite relatively few infected cells being identified by staining at 72 hpi.

Ref. [6] describes that BCoV infects the enterocytes and the positive staining of sucrase isomaltase in our study showed the enteroids contained a high proportion of these cells at the time of infection. Our results, which show a significant increase in BCoV gene copies only after the dissociation of the enteroids, indicate the need for a sufficient breakdown of the enteroids for the virus to infect the cells. The insufficient opening and, perhaps, early closing of the crypts in our study could be a reason for the relatively few infected enterocytes. According to [53], crypts from fragmented and dissociated enteroids seal and start growing from intact stem cells at the base of the crypts. It is possible that BCoV target cells are lost during fragmentation and trypsinization, resulting in mainly non-differentiated stem cells at the time of infection. For these reasons, the breakdown of enteroids might need to be optimized to increase virus access to the permissive luminal side of the cells, while minimizing the loss of cells.

In the present gene expression study, LGR5+ was upregulated in infected enteroids, which might indicate that BCoV is able to induce self-renewal of the stem cells in the crypts. A previous study on porcine intestinal organoids infected with the coronavirus transmissible gastroenteritis virus revealed upregulation of Wnt-target genes, including LGR5 with activation of the Wnt/β-catenin pathway promoting the self-renewal of intestinal stem cells [54].

As shown by the gene expression results, the BCoV-infected enteroids in our study (at least P3 at the time of infection) showed the downregulation of Muc2 at 72 hpi. Transcriptomics data from [18] comparing isolated crypts with progressive passages of organoids (up to passage 6) showed the downregulation of Muc2 at higher passage numbers. However, the absence of staining of the Muc2 indicated that any goblet cells had limited production of the Muc 2 protein. A study on porcine enteroids [27] showed that coronavirus porcine epidemic diarrhoea virus (PEDV) infected multiple intestinal epithelial cells including goblet cells. [55] demonstrated that PEDV infection reduced the number of goblet cells in the small intestine of weaned pigs. A previous study on neonatal calves also indicated BCoV infected the goblet cells, causing a reduction in the number of these cells [56]. These studies suggest that BCoV could possibly infect goblet cells, and further investigation in a well-differentiated enteroid system or in vivo would be of interest.

The expression of the immune genes IL-8 and IL-1A showed no significant difference between the mock- and BCoV-infected enteroids at 72 hpi. However, an expression study on BCoV-infected HCT-8 cells showed that both genes were highly upregulated at 24 and 72 hpi [30]. A study by [57] showed that the mean IL-8 serum concentration in BCoV-infected diarrhoeal calves increased from 0 to 24 h compared to non-diarrhoeal calves, but started to decline at 48 hpi, also suggesting that IL-8 could be used as a biomarker of intestinal injury. As these pro-inflammatory responses (IL-8 and IL-1A) are modulated as early immune responses during viral infections [58], earlier timepoints might have been more appropriate for analysis in the present study. According to [30], CXCL-3 was upregulated in BCoV-infected HCT-8 cells both at 24 and 72 hpi, whereas MMP13 expression and the TNF-α signalling pathway were upregulated at 72 hpi. In contrast, these three genes were significantly downregulated in BCoV-infected enteroids in the present study. The downregulation of MMP13 could indicate that the cells were more intact in the enteroids than in the HCT-8 cells, as MMP13 is associated with reduced epithelial barrier function during pathological changes in the cells [59]. A study by [60] showed that bovine enteroids treated with inflammatory cytokines such as TNF-α for 24–48 hpi disrupted the bovine intestinal barrier by altering the junctional morphology and permeability, thereby disrupting the epithelial cells. In contrast, an expression study on calves infected with BCoV showed the downregulation of TNF-α genes at 18 hpi [61], which corroborates with our enteroid results at 72 hpi.

Overall, the gene expression data obtained in the present study suggests that the host response to BCoV infection at 72 hpi differed substantially from that of HCT-8 cells. The discrepancies in the regulation of immune-related genes during the infection of a traditional cell line and enteroids may reflect that the enteroid system is more complex, with different cell populations that may have different modulations and can counteract proinflammatory responses. There are experimental differences between the studies, such as a much lower dose of BCoV used for the enteroids than in the HCT-8 cells, different methods to analyse gene expression (RT-qPCR and RNASeq), use of a BCoV strain that was already adapted to the cell line but not to the enteroids, and in vitro infection of cells from different host species.

## 5. Conclusions

Stem cells isolated from the ileum of a healthy calf were successfully developed into enteroids and used as a tool to increase our knowledge on host–pathogen interaction during BCoV infection. The gene expression study demonstrated a host response that differs from that reported from BCoV-infected HCT-8 cells. Future comparative gene expression studies on BCoV infection are needed to determine whether the enteroid system is a good model for the bovine intestine.

## Figures and Tables

**Figure 1 viruses-15-00635-f001:**
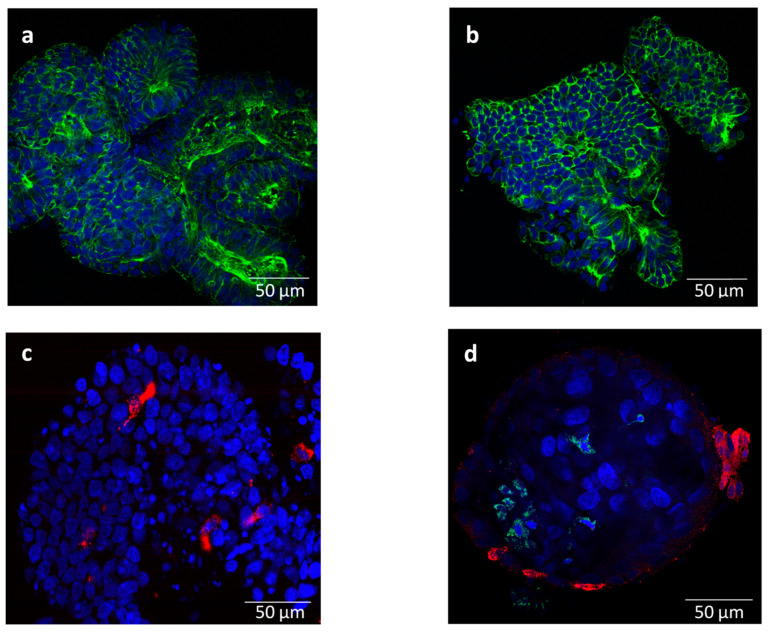
Representative immunofluorescent staining images of bovine enteroids (**a**–**c**) mock-infected and (**d**) infected with bovine coronavirus (BCoV) at 72 h post inoculation: (**a**) F-actin expressing brush border on the luminal surface (Phalloidin, green); (**b**) stained for enterocytes (sucrase isomaltase, green); (**c**) stained for enteroendocrine cells (chromogranin A, red); and (**d**) stained for BCoV S-protein (green) and enteroendocrine cells (chromogranin A, red). Nuclei were counter-stained with Hoechst (blue). Magnification = 63×; Scale bar = 50 µm.

**Table 1 viruses-15-00635-t001:** Stains and antibodies used for cell nuclei and actin, BCoV envelope S-protein, and markers for cell differentiation.

Stains/Antibodies (Producer)	Dilution	Target
Hoechst 33342 (Invitrogen, Waltham, MA, USA)	1:10,000	Nuclei
Phalloidin iFluor 488 (Abcam, Cambridge, UK)	1:1000	Actin filaments
Monoclonal mouse anti-BCoV, FITC (Bio-X Diagnostics, Rochefort, Belgium)	1:100	BCoV S-protein
Monoclonal mouse anti-sucrase isomaltase, FITC (Santa Cruz Biotechnology, Dallas, TX, USA)	1:100	Enterocytes
Polyclonal anti-bovine chromogranin A (Immunostar, Hudson, WI, USA)	1:100	Enteroendocrine cells
Goat anti-rabbit IgG, Alexa 594 (Life Technologies, Eugene, OR, USA)	1:100	
Polyclonal anti-cattle mucin 2 (MyBioSource Inc., San Diego, CA, USA)	1:100	Goblet cells
Goat anti-rabbit IgG, Alexa 594	1:100	
Polyclonal anti-cattle lysozyme (MyBioSource Inc.)	1:100	Paneth cells
Goat anti-rabbit IgG, Alexa 594	1:100	
Monoclonal mouse anti-BCoV, FITC (Bio-X Diagnostics, Rochefort, Belgium)	1:100	BCoV S-protein
Monoclonal mouse anti-sucrase isomaltase, FITC (Santa Cruz Biotechnology, Dallas, TX, USA)	1:100	Enterocytes
Polyclonal anti-bovine chromogranin A (Immunostar, Hudson, WI, USA)	1:100	Enteroendocrine cells
Goat anti-rabbit IgG, Alexa 594 (Life Technologies, Eugene, OR, USA)	1:100	

**Table 2 viruses-15-00635-t002:** Bovine coronavirus (BCoV) gene copy numbers in enteroids at 1 h and 72 h post inoculation (hpi). Samples from Setup A (fragmented enteroids) and B (fragmented and dissociated enteroids), both with *n* = 9 (3 trials with 3 parallels), were analysed by RT-qPCR using Mann–Whitney test for statistical analysis (*p* < 0.05).

Setup	Harvest Time Points (hpi)	Median BCoV Gene Copy Number	IQR	*p*-Value
A	1	2967	1877–5973	0.3409
	72	3723	1150–6047	
B	1	2962 *	2298–4741	<10^−5^
	72	20,956 *	12,646–32,889	

* Statistically significant difference (*p* < 0.05).

**Table 3 viruses-15-00635-t003:** Gene expression ratios for selected genes in BCoV- to mock-infected bovine enteroids. The enteroids were harvested 72 h post inoculation and mRNA was quantified by two-step RT-qPCR. The results were normalized to housekeeping genes GAPDH and ACTB. Three independent trials with mock- (*n* = 9) and two with BCoV-infected (*n* = 6) enteroids are included.

Gene	Type	Reaction Efficiency	Expression Ratio	Std. Error	95% C.I.	P(H1)	Result
*GAPDH*	REF	0.99	1.127				
*ACTB*	REF	1.01	0.887				
*LGR5*	TRG	1.01	2.811	0.915–5.705	0.554–6.948	0.022	UP
*ChrA*	TRG	1.02	8.356	3.292–18.151	2.046–42.257	0.000	UP
*Muc2*	TRG	1.09	0.213	0.052–0.835	0.019–2.676	0.008	DOWN
*IL-8*	TRG	0.95	1.110	0.675–1.782	0.532–2.399	0.537	No diff
*IL-1A*	TRG	0.94	1.091	0.653–1.733	0.417–2.348	0.662	No diff
*MMP13*	TRG	1.09	0.036	0.012–0.145	0.010–0.372	0.000	DOWN
*CXCL-3*	TRG	1.1	0.554	0.287–0.848	0.240–1.031	0.009	DOWN
*TNF-α*	TRG	1.03	0.556	0.336–0.891	0.235–1.108	0.004	DOWN

P(H1): Probability of alternate hypothesis that the difference in gene expression between BCoV- and mock-infected enteroids is due only to chance. REF: Reference (housekeeping gene). TRG: Target gene. Geometric mean of expression ratio of housekeeping genes = 1.00.

## Data Availability

Full genomes of BCoV from several samples used in this study were sequenced and can be accessed with GenBank accession ID OQ507475.

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
