# Peer review of "Bovine Enteroids as an In Vitro Model for Infection with Bovine Coronavirus"

_viruses, 2023, doi:10.3390/v15030635_

Round 1
Reviewer 1 Report
Comments to the Author
The manuscript entitled " Bovine enteroids as an in vitro model for infection with bovine coronavirus" provides insight in preparation of 3D model for studying host-pathogen interactions. The study is precisely designed with an important goal to provide in vivo 3D replication system for BCoV, where gene expression of 8 different genes was compared. Obtained results are precisely statistically evaluated.
The manuscript is very well written. Some minor suggestions for improving the manuscript are indicated below and also in the appendix.
1. In the Introduction: Please add short description of expression of genes used in your study in the introduction.
2. In the Introduction: This statement could mislead the reader. Could you explain that: BCoV is the cause of 3 distinct clinical syndromes in cattle: calf diarrhea (CD), winter dysentery (WD) with hemorrhagic diarrhea in adults, and respiratory infections in cattle of various ages including the bovine respiratory disease complex (BRDC).
3. In section2.3: I suggest adding more information on strain used. I am sure that BCoV strain used in your study was classified by sequencing, therefore I suggest adding accession number of sequence, if possible.
4. In section 2.5: Please add reference (35) and also information that you used already published RT-qPCR protocol for the detection of BCoV.
5. In section 2.7: Please add producer after the reagent (DNAse I). Also consider using uniform citing for all producers.
6. In sections 2.6 and 2.7: I suggest expressing efficiency in %.
7. The Discussion part is very complex, but it is precisely written with a lot of comparisons with the literature. Authors also describe the limitations of their study and provide very promising conclusion on using enteroids as a tool to study characteristics of BCoV infection.

Author Response
- In the Introduction: Please add short description of expression of genes used in your study in the introduction.
A short description has been added in the manuscript as per the suggestion in line 69-73.
- In the Introduction: This statement could mislead the reader. Could you explain that: BCoV is the cause of 3 distinct clinical syndromes in cattle: calf diarrhea (CD), winter dysentery (WD) with hemorrhagic diarrhea in adults, and respiratory infections in cattle of various ages including the bovine respiratory disease complex (BRDC).
The sentence has been replaced as suggested by the reviewer.
- In section 2.3: I suggest adding more information on strain used. I am sure that BCoV strain used in your study was classified by sequencing, therefore I suggest adding accession number of sequence, if possible.
We have sequenced full genomes of BCoV from several samples, however, the sequences have not been published and can be provided on reasonable request. Generally, the genomic variation between BCoVs is limited.
- In section 2.5: Please add reference (35) and also information that you used already published RT-qPCR protocol for the detection of BCoV.
The sentence has been modified to, “Information on primers, probe, and RT-qPCR conditions is provided in table S2. RT-qPCR for BCoV was performed based on a published protocol [38] using the RNA UltraSense™ One-Step Quantitative RT-PCR System kit (Invitrogen, MA, USA).”
- In section 2.7: Please add producer after the reagent (DNAse I). Also consider using uniform citing for all producers.
Producer name has been added and corrected throughout the manuscript.
- In sections 2.6 and 2.7: I suggest expressing efficiency in %.
Efficiency has now been expressed in %.
Reviewer 2 Report
This is a very nicely presented manuscript describing the establishment and optimisation of bovine enteroids, and their ability to support infection by a bovine coronavirus.
The introduction and discussion are thorough, exploring data from other published work, not only with bovine systems, but other species as well. The figures and legends are clear and comprehensive. The conclusions are clearly supported by the data presented.
Specific comments:
Section 2.1, first paragraph - Was a home office license required for the work, please provide details if so, or the reason why not.
Section 2.5, first paragraph - Please add detail about the sample preparation for RNA extraction (amounts used, lysis method, etc.)
Author Response
Section 2.1, first paragraph - Was a Home Office license required for the work, please provide details if so, or the reason why not.
The following sentence has been added in Institutional Review Board Statement, "Tissues used were obtained from post-mortem of a healthy male British Holstein–Friesian (Bos taurus) calf and therefore did not require Home Office license." And, in section 2.1 the ethical committee reference number has been added.
Section 2.5, first paragraph - Please add detail about the sample preparation for RNA extraction (amounts used, lysis method, etc.)
The enteroids were harvested in 300 μl of RLT buffer plus DTT (2M) Qiagen as described in section 2.3 and mixed well by pipetting.